Layered patterns in nature, medicine, and materials: quantifying anisotropic structures and cyclicity

Smolyar Igor 1 igorsmolyar8755@gmail.com
http://orcid.org/0000-0002-9843-7993 Bromage Tim 2
Wikelski Martin 3
1 National Centers for Environmental Information, National Oceanic and Atmospheric Administration , Ashvelle, NC , USA
2 Department of Biomaterials & Biomimetics and Basic Science & Craniofacial Biology, College of Dentistry, New York University , New York City, NY , USA
3 Max-Planck Institute for Ornithology and Department of Biology, Konstanz University , Radolfzell and Konstanz , Germany
Gollo Leonardo
Electronic publication date: 2019 Oct 14
Publication date: 2019
Volume: 7
Electronic Location ID: e7813
Received 2019 Feb 11; Accepted 2019 Sep 2
Copyright: © 2019 Smolyar et al.
Copyright year: 2019
Copyright holder: Smolyar et al.
License: This is an open access article distributed under the terms of the Creative Commons Attribution License, which permits unrestricted use, distribution, reproduction and adaptation in any medium and for any purpose provided that it is properly attributed. For attribution, the original author(s), title, publication source (PeerJ) and either DOI or URL of the article must be cited.
License URL: https://creativecommons.org/licenses/by/4.0/

Keywords: Anisotropy of layered systems, Boolean functions, Structural anomaly, 0-gravity, N-partite graph, World ocean, Disorder of structure, Anisotropic growth

Funding: The authors received no funding for this work.

==============================
Various natural patterns—such as terrestrial sand dune ripples, lamellae in vertebrate bones, growth increments in fish scales and corals, aortas and lamellar corpuscles in humans and animals—comprise layers of different thicknesses and lengths. Microstructures in manmade materials—such as alloys, perlite steels, polymers, ceramics, and ripples induced by laser on the surface of graphen—also exhibit layered structures. These layered patterns form a record of internal and external factors regulating pattern formation in their various systems, making it potentially possible to recognize and identify in their incremental sequences trends, periodicities, and events in the formation history of these systems. The morphology of layered systems plays a vital role in developing new materials and in biomimetic research. The structures and sizes of these two-dimensional (2D) patterns are characteristically anisotropic: That is, the number of layers and their absolute thicknesses vary significantly in different directions. The present work develops a method to quantify the morphological characteristics of 2D layered patterns that accounts for anisotropy in the object of study. To reach this goal, we use Boolean functions and an N-partite graph to formalize layer structure and thickness across a 2D plane and to construct charts of (1) “layer thickness vs. layer number” and (2) “layer area vs. layer number.” We present a parameter disorder of layer structure (DStr) to describe the deviation of a study object’s anisotropic structure from an isotropic analog and illustrate that charts and DStr could be used as local and global morphological characteristics describing various layered systems such as images of, for example, geological, atmospheric, medical, materials, forensic, plants, and animals. Suggested future experiments could lead to new insights into layered pattern formation.

Introduction

Layered structures can be found in various patterns—including satellite images of the surfaces of Mars, Pluto (Fig. 1A), Titan (Fig. 1B), in lamella bones (Fig. 1C), human aorta (Fig. 1D), and butterfly wings (Fig. 1E). Snake and spider skin, coral growth increments, leaf structures and flower surface microstructures, alloy (Fig. 2A), fish skin (Fig. 2B), wild turkey wings (Fig. 2C), bird plumage patterns, three-dimensional (3D) images of shells (Fig. 2D), lightning (Fig. 2E), human and animal hairs, and skeletal muscle (Fig. 2F) all exhibit patterns of this type.

Figure 1 Living and non-living layered systems.

(A) Layered structure of surface of Pluto. (B) Layered structure of surface of Titan. (C) Cross-section of lamellar bone. (D) Human aorta, an example of soft layered tissue. (E) Layered structure constructed from Morpho butterfly wing scales. Image credit: (A) NASA/JHUAPL/SwRI. (B) NASA/JPL. (C) Norman Barker. (D) Hans Snyder/JMD Histology & Histologic Inc. (E) Didier Descouens. Wikipedia contributor. “Morpho.” https://en.wikipedia.org/wiki/Morpho#/media/File:Morpho_didius_Male_Dos_MHNT.jpg licensed under CC BY SA 4.0.

Figure 2 3D and 2D layered systems.

(A) Steel lamellar microstructure consists of pearlite in a ferrite matrix. (B) Fish skin layered camouflage (C) Wild turkey feathers as a building block of anisotropic layers. (D) 3D anisotropic layered structure of an abalone shell. (E) Cloud-to-ground lightings as an example of atmosphere phenomena with anisotropic layered structure. (F) Pattern of human skeletal muscles as an example of soft tissue with anisotropic layered structure. Image credit: (A) Michelshock. Wikipedia contributor. “Pearlite.” https://commons.wikimedia.org/wiki/File:Pearlite.jpg. Public domain. (B) Leonard Low. Wikipedia contributor. “Triggerfish,” https://en.wikipedia.org/wiki/Triggerfish#/media/File:Balistapus_undulatus.jpg, licensed under CC BY 2.0. (C) Moscow Hide and Fur (www.hideandfur.com). (D) Sylvain Deville. (E) C. Clark. NOAA Photo Library/NOAA Central Library; OAR/EPL/National Severe Storm Laboratory. (F) Norman Barker.

Natural layered patterns are attractive objects of study for specialists of different disciplines for several reasons. First, layer thickness and structure represent the cumulative effect of internal and external factors regulating pattern formation. Thus, layered patterns serve as a record of diverse events occurring in different space–time domains. This record makes it possible to link the morphology of layered patterns to external factors such as variability in the Earth’s rotation (Pannella, 1972), climate cycles (Radebaugh et al., 2011; Ewing, Hayes & Lucas, 2014), and the state of the environment (Guyette & Rabeni, 1995; Costa, Pereira & Oliveira, 2002).

Some soft tissues—including the human aorta and Pacinian (lamellar) corpuscles (PC)—exhibit layered structures. PC are nerve endings in the skin responsible for detecting and locating skin deformations produced by air vibrations and skin contact (Kaas, 2012). Studying their morphological parameters has implications for the development of new technology for conveying speech and visual information through somatosensory channels (Biswas, Manivannan & Srinivasan, 2015). The human aorta has a layered (i.e., lamellar) structure that typifies the elastic lamina found in human and animal blood vessels. The study of aortic microstructure and age-related changes is an urgent area of medical research (Novotny et al., 2017; Selçuk et al., 2015).

Additionally, analyzing layer morphology is an essential element of solving many problems in materials science, biomimetic, and forensic research. For instance, biometric research has explored the structural properties of butterfly photonic systems (Vukusic & Sambles, 2003), flower surfaces (Barthlott, Mail & Neinhuis, 2016; Huang, Hai & Xie, 2017), and snake skin (Abdel-Aal & Mansori, 2011; Klein & Gorb, 2012; Filippov & Gorb, 2016). In materials science, a material’s mechanical and physical characteristics are determined by its microstructure, which in many instances is layered (Moya, 1995; Mayer, 2005). Understanding the relationship between microstructure and these properties is vital for developing porous materials with new mechanical characteristics (Deville, 2018). In forensic research, morphological features of layered systems in hair and fingerprints can be used for identification purposes (Champod, 2015; Lee et al., 2014).

The study and commercial applications of various categories of layered systems requires formalizing aspects of their analysis. One of the first steps toward this goal is quantitatively describing the morphology of a layered pattern. Formalizing this morphology is problematic because of the numerous breaks and confluences (i.e., bifurcations) in the layers of two-dimensional (2D) objects (Blumberg, 2006). The number and thickness of these layers is a function of the direction in which they are measured; that is, they are anisotropic in both size (including thickness and area) and structure, thereby making it difficult to develop a formal procedure for their analysis.

To address this problem, we have proposed an empirical model (EM) of 2D layered patterns, with the aim of providing tools to quantify the morphological features of anisotropic layered objects (Smolyar, Ermolaeva & Chernitsky, 1987; Smolyar & Bromage, 2004; Smolyar, 2014; Smolyar, Bromage & Wikelski, 2016). EM has three components: an N-partite graph (G(N)) (Fig. 3A), a Boolean function (BF) (Figs. 3B–3D) to describe the 2D structure of layers, and Table TM,N, which comprises the thickness of layers along transects R1, …, RN; EM = {BF, G(N), TM,N}. Transects R1, …, RN are straight lines always distributed evenly across a 2D layered pattern. The concept of open/closed gates (Figs. 3B and 3C) makes it possible to describe all possible versions of layer structure using BFs (Fig. 3D).

Figure 3 Structure formalization of layered patterns.

(A) Description of structure of a binary anisotropic pattern in terms of N-partite graph. (B) “Gate open” and “gate close.” (C) Example of layer structure for two states of gates. (D) Structure of layers as a function of gate states for fragments of pattern shown in (B).

The procedure for converting a binary layered pattern into EM consists of two basic elements: First, we convert the binary pattern in its raster (i.e., pixel) format into a binary array in comma separated value (CSV) format. In CSV, white pixels have a value of 0 and black pixels have a value of 1 (Fig. 4A). XY elements of the binary array represent the XY coordinates of the pixel on the image of the 2D plane. Second, we segment and label the resulting binary patterns, assigning to each continuous line a unique identification number (ID). This procedure allows us to calculate the coordinates of intersection points of transects with layers and calculate layer thickness along N transects (Fig. 4B). In this case, black pixels are designated as the foreground of the pattern and white pixels as background. In order to construct G(N), black pixels are designed as background and white pixels as foreground (Fig. 4C). The procedure for constructing EM is described in detail in Smolyar, Bromage & Wikelski (2016) and Smolyar (2014).

Figure 4 Basic elements of empirical model of layered patterns.

(A) Binary layered image in Comma Separated Value format. (B) Comma Separated Value format as the source of layer thickness. (C) Comma Separated Value format as the source of layer structure.

Layered systems—irrespective of their nature or size—share several key elements, including the idea of layers, which have thicknesses and length. EM provides tools to formalize these elements, which are defined in terms of transects that cross a pattern from its lower to its upper margins (Fig. 3A). We introduce the concept of synchronizing layer formation across a 2D plane in order to quantify the structure of layers and develop a procedure for plotting (1) “layer thickness vs. layer number” and (2) “layer area vs. layer number” (Smolyar, Bromage & Wikelski, 2016). That is, to construct the structure of each layer across a 2D plane, it is necessary to synchronize layer formation in the space–time domain. Because layers are anisotropic, more than one version of the layered structure could be used for synchronization, resulting in fuzziness in the charts for “layer thickness vs. layer number” and “layer area vs. layer number.” Fuzziness is an unavoidable attribute when parameterizing anisotropic layered patterns. When describing the variability of layer size in anisotropic patterns across a 2D plane, high accuracy and high confidence are mutually exclusive.

Smolyar, Bromage & Wikelski (2016) introduced the idea of an “index of confidence,” which allows a compromise between detail and signal-to-noise ratio—either more detail and a lower signal-to-noise ratio or less detail and a higher signal-to-noise ratio—when describing the variability of layer thickness and area across N transects. It is therefore possible to plot robust charts for “layer thickness vs. layer number” and “layer area vs. layer number.” These charts describe the global morphological characteristics of an entire 2D layered pattern. For instance, if each layer (e.g., of tree rings, fish scales, lamellar bones, corals) is associated with the instant of time tj in which it was formed, then a layer’s thickness and area are measures of the growth rate of the layered system at that time. In this case, “layer thickness vs. layer number” and “layer area vs. layer number” are interpreted as growth-rate variability across the entire system of 2D anisotropic layers.

Using EM to analyze the growth-rate variability of lamella bones allows us to reveal cyclicity in bone formation not previously observed (Bromage et al., 2009). These results—as well as evidence that many factors controlling pattern formation are cyclic in nature—motivate us to use EM to reveal and quantify cyclicity for layered objects when layer formation is not associated with a moment of time, tj.

The present paper continues our previous work (Smolyar, Bromage & Wikelski, 2016). The goals of the paper are twofold: (1) develop a method for quantifying the structural characteristics of layered patterns and (2) examine the applicability of structural characteristics and EM = {BF, G(N), TM,N} for analyzing layered patterns of various categories. To reach these goals, wereview layered patterns appearing in the realms of medicine, forensics, geology, plants, animals, and materials science in order to justify that similarities in the structural anisotropy of layers can be described by EM;

introduce a structural characteristic of layered patterns called “Disorder of layer structure” (DStr) and propose a fully automated method for its calculation. DStr serves as a measure of deviation from an isotropic analog in patterns with anisotropic layered structure;

illustrate that DStr is a universal characteristic applicable to any 2D layered pattern, irrespective of nature and size, and could be used as a local and global defining characteristic of a layered pattern;

illustrate the possibility of using an EM of layered patterns, to quantify the variability of layer thickness across 2D planes of images of objects of various categories.

Various examples underline the applicability of DStr and EM for quantifying the structural characteristics of various categories of living and non-living layered systems.

We justify the usefulness of the proposed metric by showing that DStr could be used toreveal structural anomalies in layered patterns;

monitor structural changes in sand dunes/ripples over period of time on the surface of Earth and Mars;

formulate testable hypotheses by setting correspondence between physical, mechanical, and biological properties of objects under study and the morphological characteristics of their layered patterns.

We also give suggestions for further experiments that have the potential to help us better understand environmental influences on pattern formation.

In different publications, layers may be called growth lines (Kahn & Pompea, 1978), circuli (Izzo & Zydlewski, 2017), bands (Smolyar & Bromage, 2004), growth increments (Carroll et al., 2014, Goodwin et al., 2001), lamellae (Biswas, Manivannan & Srinivasan, 2015), dunes (Blumberg, 2006), ripples (Calvani et al., 2016), or ridges (Wilson & Zimbelman, 2004), depending on the object of study. The present work uses these terms synonymously.

Method

This section explains DStr and describes two structural extremes of anisotropic 2D layered systems: minimal disorder (DStr = 0) and maximal disorder (DStr = 1). Also, we justify the way in which DStr allows layered patterns to be distinguished from non-layered patterns.

Basic concept

A precise definition of anisotropy or isotropy depends on the object of study. Our definition of isotropic and anisotropic layered patterns comes from the study of growth increments in fish scales. Growth rates of fish scales vary in different directions, resulting in numerous breaks and confluences in growth layers, which are the source of anisotropy in fish scale growth layers because more than one possibility exists for describing layer structure across a 2D plane (i.e., across N transects). In other words, the structure of layers is a function of the state of gates (Figs. 3B–3D). Therefore, characteristics of anisotropy in a layered pattern are (i) the possibility of more than one version of layer structure and (ii) different lengths of layers, where length is defined as the number of transects crossing the layer. In an isotropic image, each layer is crossed by all N transects (i.e., layers have no breaks and confluences), and only one possibility exists to describe the structure of each layer. Objects with isotropic layered structure are relatively rare. Hence, a general definition of anisotropy implies different properties in various different directions, and anisotropy in a layered system implies different properties in the directions of layer formation only.

Two-dimensional layered patterns consist of both isotropic and anisotropic components. We therefore define the DStr of a 2D layered pattern as the measure of a pattern’s deviation from isotropy. Because the N-partite graph, G(N), represents the structure of a layered pattern, isotropic and anisotropic components could be understood in terms of edges and vertices in G(N).

G(N) consists of a sequence of bi-partite graphs, G(R1, R2), … G(Rj, Rj+1), … G(RN−1, RN), where G(Rj, Rj+1) is a bi-partite graph that describes the structure of a layered pattern situated between transects Rj and Rj+1 (Figs. 5A and 5B). An isotropic layer here would imply that vertex a ∈ Rj connects only with vertex b ∈ Rj+1 and b ∈ Rj+1 connects only with a ∈ Rj. Edge ab in G(Rj, Rj+1) is therefore an isotropic edge. TotalEdges denotes the total number of edges in G(Rj, Rj+1). The number of anisotropic edges in G(Rj, Rj+1) is equal to TotalEdges minus the number of isotropic edges.

Figure 5 Quantifying disorder of layer structure (DStr) in an anisotropic layered pattern.

(A) Layer structure for three transects: R1, R2, and R3. (B) Layer structure for four transects: R1, R2, R3, and R4. (C) Disorder of layer structure as a function of layer number. (D) Disorder of layer structure as a function of relative layer number.

Disorder (DGrp (Rj, Rj+1)) of bi-partite graph G(Rj, Rj+1): DGrp(Rj,j+1)=numberofanisotropicedgesTotalEdges

Disorder of N-partite graph G(N): (1) DGrp(R1,RN)=1N−1∑j=1N−1⁡(DGrpj,j+1)

Two questions follow from Eq. (1):

Question #1. From Eq. (1), it transpires that DGrp(R1, RN) depends on sampling density (i.e., the number of transects used to calculate DGrp(R1, RN)). How many transects should be used to quantify DGrp(R1, RN), which has not yet been technically defined?

Question #2. Following Eq. (1), DGrp(R1, RN) varies from 0 to 1. If DGrp(R1, RN) = 0, then the layered pattern is entirely isotropic; such layered images are easily visualized. But what do entirely anisotropic patterns (that is, DGrp(R1, RN) = 1) look like?

The following two subsections address these questions.

Sampling density

Because the anisotropic components of a layered pattern are unevenly distributed in 2D space, we examine multiple versions of sampling density to determine how many transects are necessary to quantify DStr. We plot the function y = f(x) (i.e., DStr = f(number of transects)), which describes dynamic changes in DGrp when the number of transects tends to the maximum possible number. The area bounded by y = f(x) and the axis y = 0 is the measure of DStr.

The choice of how many transects, R1, …, Rj, …, RN, to use to develop the EM plays an essential role in analyzing the structure of anisotropic layered patterns. Consider the proposed approach for constructing sets of transects used to describe BF, G(N), TM,N, and calculate DStr.

The general principle in choosing the number of transects is based on the fact that we do not know a priori how many transects will best describe the particular layered pattern within the frame of finding a solution to the specific problem. In these circumstances, our choice is to examine as many different versions of transect sets. In the present work, all transects are straight lines, and the distance between two adjacent transects remains constant across all transects.

Figure 5 illustrates the procedure for constructing y = f(x) and calculating DStr. We calculate DStr for 100 versions of transect sets. The first version consists of three transects (N = 3, Fig. 5A) and the second of four transects (N = 4, Fig. 5B). The last version consists of 103 transects (N = 100). Figure 5C shows equation DStr = f(number of transects). The distance between any two nearby transects is equal to 1 pixel for N = 100. We calculate DStr for a normalized number of transects Ni(normalized)= Ni/Nmax in order to present the results of calculating DStr in scale (0,1) (Fig. 5D). In this case, y = f(x) contains as much structural detail as possible for the layer pattern under study.

We refer to the number of transect sets used to plot y = f(x) and calculate DStr as “sampling density.” Sampling density is “highest” if all possible versions of transect sets are used to construct y = f(x) and calculate DStr (Fig. 5D). Sampling density could be described as “medium” or “low” depending on the number of transect sets used to construct y = f(x). An experiment with the pattern of a human hair in the Results section illustrates how high, medium, and low sampling density affect the shape of y = f(x).

The coefficient of determination, R2 (Draper & Smith, 1998), ranges from 0 to 1 and is used to estimate how well the partial-rational function y = mxk replicates y = f(x). If R2 = 1, then y = mx−k is the approximation of y = f(x) with 0 error. We choose function y = mx−k to replicate y = f(x) for two reasons. First, it contains two numeric coefficients, m and k, so only two numeric values serve as global structural characteristics of the entire 2D layered pattern. Second, for many layered patterns, R2 ≥ 0.90 for y = mx−k.

We use Microsoft Excel 2007 to calculate parameters m and k for y = mx−k and R2 for y = mx−k. Because R2 = 0.9962, y = 0.0228x−0.969 (Fig. 5D); thus, Eq. (2) can be used to calculate DStr:(2) DStr=∫01⁡f(x)dx

For the pattern in Fig. 4D, DStr = 0.08346.

Maximal structural disorder of layered patterns (DStr = 1)

Consider the appearance of a layered pattern with DStr = 1 (i.e., the layered pattern’s has no isotropic components): Each vertex situated along transect Rj connects with all vertices situated along Rj+1; thus, the bi-partite graph G(Rj, Rj+1) is complete. If the layered image consists of complete bi-partite graph sequences for all possible numbers of transects, then DStr = 1. It should be stressed that we do not use isolated vertices (i.e., those that are not connected to other vertices) in calculating DStr, because they do not form isotropic or anisotropic edges. One possible example of a pattern in which DStr approaches maximal structural disorder is stars in the night sky (Smolyar, Bromage & Wikelski, 2019: Fig. 11).

Layered vs. non-layered systems

It is transparent that the transition from layered to non-layered patterns occurs continuously and monotonously, which raises the question of whether it is possible to distinguish between layered and non-layered images. Let us consider how we can use DStr to answer this question. DStr could be defined in either of two ways:DStr is the area between y = 0 and the function y = f(x) (Fig. 5D). In this case, DStr is the measure of deviation of an anisotropic pattern from isotropy, denoted by DStr(deviation from isotropy).

DStr could be interpreted as the deviation of an anisotropic pattern from a system with maximal disorder (i.e., a chaotic system), which is defined as the area between y = f(x) and y = 1, denoted DStr(deviation from chaos).

Thus, y = f(x) divides a 1 × 1 square into two areas (Fig. 5D):

DStr(deviation from isotropy) and DStr(deviation from chaos).

Because the area of the square is equal to 1, thus(3) DStr(deviationfromisotropy)+DStr(deviationfromchaos)=1

Using Eq. (3), it is possible to quantitatively describe a layered pattern in the following manner: If DStr ≤ 0.5, then the structure of a pattern is more layered then chaotic; if DStr > 0.5, then the structure is more chaotic than layered. Therefore, DStr ≤ 0.5 is the maximal possible value for the characteristic of disorder in the structure of layered patterns. This is why a threshold of 0.5 is used to describe the difference between the structures of layered patterns in percentages.

Results

This section presents the results of calculating DStr, the variability of layer size across a 2D plane, and experiments illustrating the sensitivity of these methods to detecting minor changes in layered structures.

Image binarization

Image binarization have served for decades as a powerful tool for solving a broad spectrum of problems across various disciplines (Sezgin & Sankur, 2004; Tensmeyer & Martinez, 2017), but “there is no way to select single or best method which is used for all images” (Garg & Garg, 2013).

The central element of a binarization procedure is the threshold that allows objects to be separated from the background of a grayscale image. Binarization can be classified into two basic categories—local and global—based on the goal of the image analysis and its specific characteristics (Ekhande, Rumane & Ahire, 2015). Local methods are based on finding a threshold for each individual pixel or local area of an image; global methods attempt to find a single threshold for the entire image.

Several factors define our particular approach to binarization. First, we have the freedom to choose images for the present work, allowing us to select images with equal illumination and uniform backgrounds and without imaging artifacts. Second, a distinctive feature of layered images is that high image gradient indicates the edge of a layer. In this case, image embossing allows us to detect the edges of layers (Prajapati & Shah, 2017). Experiments in binarizing layered patterns show the effectiveness of the emboss operation to distinguish distinct layers from an image background (Smolyar, Bromage & Wikelski, 2016), resulting in the choice of one threshold for an entire image. Thus, the essential steps for binarization are image embossing and using a single threshold to assign each pixel a value of 0 or 1.

We also investigate how two distinct modes of image vectorization (contour trace and central line trace) effect DStr. To do so, we first embossed an image, used contour trace and central line trace modes to present the image in the binary vector format and then saved the image in raster binary format. The DStr parameters are calculated for contour trace and central line trace images. Outcomes of these experiments are presented in the section Sensitivity of DStr to binarization of layered patterns.

Image preprocessing

The algorithm for calculating DStr consists of the following steps:The original layered image (in grayscale raster format) is converted into EM using the technology described in Smolyar (2014) and Smolyar, Bromage & Wikelski (2016).

Transects R1, …, Rj, …, RN are plotted and DGrp(Rj, Rj+1) is calculated (Eq. (1)).

Step 2 is repeated P times, resulting in DGrp(1, N1), DGrp(1, N2), …, DGrp(1, NP).

The function y = f(x) is constructed and R2 is calculated.

DStr for the entire sampling area is calculated (Eq. (2)).

Disorder of layer structure (DStr)

Geology. Figure 6A depicts structural anomalies in a layered system of the Martian surface. The DStr of the sampling area a is 10 times less than that of nearby area b, which exhibits structural anomaly with respect to area a. Figure 6B depicts the structural anomaly of sand ripples. The DStr of the orange and blue areas is 2.3 times less than that of the red area. There is a 33% difference in DStr between areas a and b (Fig. 6A) and a 55% difference in DStr between the red and the blue/orange sampling areas (Fig. 6B). DStr provides a tool to reveal structural changes in dunes/ripples patterns over a period of time on the surfaces of Marth and Earth.

Figure 6 Morphological characteristics of structural anomalies.

(A) Region with natural anomaly in layered system. Two nearby Martian regions, a and b, with substantially different levels of structural disorder. (B) Region (red rectangle) with anomalies in sand dune layer structure caused by animal migration. Image credit: (A) NASA/JPL/Univ. of Arizona. PSP_008641_2105. (B) Martin Harvey.

Figures 6A–6B is an example of structural anomalies in an anisotropic layered system. Such anomalies could exist in any category of object of study. For instance, structural anomalies in metal microstructures could be interpreted as cracks.

Materials. The morphology of the surface of black diamonds is an essential element in developing solar energy conversion systems (Calvani et al., 2016). Figures 7A–7C shows the dynamic of DStr as a function of different treatments dose of the black diamond surface. Thus, it is potentially possible to describe the relationship between the structural characteristics of a black diamond’s surface (Fig. 7D) and treatment doze in quantitative terms (i.e., construct function DStr = f(D), where D is the notation for the treatment doze). Equation DStr = f(D) is a step toward identifying links between material microstructures and their physical properties.

Figure 7 Disorder of layer structure in black diamonds as a function of the treatment dose (D).

(A) Microstructure of black diamond for D = 2.5 kJ cm2. (B) Microstructure of black diamond for D = 5.0 kJ cm2. (C) Microstructure of black diamond for D = 12.5 kJ cm2. (D) Morphological characteristics of black diamond surface. Image credit: Calvani et al. (2016).

Medicine. The pattern of the human aorta (Fig. 8A) is an example of a layered pattern with complicated structural anisotropy. Four segments—blue, orange, green, and red—show similar but not identical DStr (Fig. 8B). There is a 4% difference between DStr(orange) and DStr(green), which is probably noise from converting the initial color image to black and white. It is necessary to point out that DStr is the result of averaging DGrp(R1, RN) for different versions of transect numbers (Eq. (2)). This does not necessary mean that for some version of transect numbers the differences between DStr of comparable segments can’t be substantially greater of the averaged version of DStr. Figures 8C–8D shows that for a version of transects equal 0.1, the difference between DStr(blue) and DStr(green) is 0.12, a 26% difference. Thus, it is possible to assume that 26% is above noise and there are some structural differences between green and blue aorta segments.

Figure 8 Disorder of layer structure in lamellar human aorta.

(A) Binary image of human aorta divided into four regions. (B) Parameter DStr for four regions. (C) Charts DStr = f(number of transects) for two regions with maximal difference in DStr. (D) Relative number of transects equal 0.12 maximized differences in DStr between blue and green regions of human aorta. Image credit: Hans Snyder/JMD Histology & Histologic, Inc.

Birds. Plumage patterns in banded pitta, kingfisher, harpactes-1, harpactes-2, and owl (Figs. 9A–9G) offer examples of layered systems in bird plumage. DStr shows significant diversity in the structure of these layered systems: DStr(giant kingfisher) = 0.7591 (Fig. 9A); DStr(harpactes-2) = 0.165 (Fig. 9E). This result inspires us to ask whether parameters of layered structures might serve as phenotypic characteristics (Gluckman & Mundy, 2016). EM could be used to examine the structural characteristics of birds’ plumage patterns with respect to state of the environment.

Figure 9 Disorder of layer structure in bird plumage patterns.

(A) Sampling area (white rectangle) of giant kingfisher. (B) Sampling area (white rectangle) of banded pitta. (C) Sampling area (white rectangle) of polar owl. (D) Sampling area (white rectangle) of harpactes 1. (E) Sampling area (white rectangle) of harpactes 2. (F) Morphological characteristics of bird plumage patterns. (G) Disorder of bird plumage patterns as function of number of transects. Image credits: (A) Charles J. Sharp. Wikipedia contributor. “Giant kingfisher,” https://en.wikipedia.org/wiki/Giant_kingfisher#/media/File:Giant_kingfisher_(Megaceryle_maxima)_male.jpg, licensed under CC BY SA 4.0. (B) Doug Janson. Wikipedia contributor. “Javan banded pitta,” https://en.wikipedia.org/wiki/Javan_banded_pitta#/media/File:Pitta_guajana-20030531.jpg CC BY SA 3.0. (C) Phillip Anderson. (D) J.J. Harrison (https://en.wikipedia.org/wiki/Orange-breasted_trogon#/media/File:Harpactes_oreskios_-_Kaeng_Krachan.jpg), licensed under CC BY SA 3.0.Wikipedia contributor. “Harpactes.” (E) Lip Kee. Wikipedia contributor. “Harpactes,” https://en.wikipedia.org/wiki/Scarlet-rumped_trogon#/media/File:Scarlet-rumped_Trogon_(Harpactes_duvaucelii)_-_Flickr_-_Lip_Kee_(2).jpg, licensed under CC BY SA 2.0.

Forensic (human hair). Figures 10A–10B shows that high sampling density accounts for more structural details than low sampling density. Thus, sampling density allows us to quantify as many structural details as resolution of an object of study permit.

Figure 10 Disorder of layer structure as function of sampling density.

(A) Image of a human hair. (B) High sampling density shows more structural details in layered human hair patterns than low and medium sampling density. Image credit: Russ Crutcher. www.microlabgallery.com.

Forensic (fingerprints). As Fig. 11A indicates, the four basic categories of fingerprint patterns have distinctive structural characteristics that vary from DStr(plain arch) = 0.065 to DStr(central pocket loop) = 0.143. Distinctions between DStr among different categories of fingerprints substantially depend on the number of transects used to calculate DStr. To define the number of transects that allow maximal differences between plain arch and central pocket loop fingerprints, we plot the chart (Fig. 11B) as DStr(central pocket loop) − DStr(plain arch) = f(number of transects). A total of 15 transects allow the maximal possible differences between two categories of fingerprints; that is, DStr(central pocket loop) − DStr(plain arch) = 0.2, which is 2.56 times more than the DStr(central pocket loop) and DStr(plain arch) comparison if Eq. (2) is used to calculate DStr. Sampling density and number of transects could complement DStr in forensic identification.

Figure 11 Disorder of layer structure of fingerprints of four categories.

(A) Four basic categories of fingerprints and their morphological characteristics. (B) Comparison of DStr for fingerprint images of central pocket loop and plain arch.

An Excel file (File S1) presents raw data for calculating DStr = f(transect number) and DStr. For instance, data for Fig. 11 can be found in the Excel sheet titled Fig. 11. The column labeled “Transect #” indicates the number of transects used to calculate DStr. The parameter DStr is equal to the sum of the areas of trapezoids formed by the chart of y = f(x).

Cyclic variability of layer size across 2D plane

The algorithm for constructing a chart for layer thickness vs. layer number is identical to that used in Smolyar, Bromage & Wikelski (2016). The experiments described in this section examine the distribution of layer thickness across a sampling area in order to estimate whether average layer thickness accurately describes the morphological characteristics of 2D layered systems.

Geology. Dune fields are an example of the layered patterns that exist throughout nature. Dune spacing (i.e., layer thickness) is a basic morphological characteristic of dune systems (Lancaster, 2009). Figures 12A–12C shows layered fragments of the surface of Mars that have isotropic structure (i.e., all fragments have DStr = 0), which allows us to describe the variability of layer thickness across the 2D sampling area with high accuracy. Several transects are used to calculate average thickness of each layer. Charts of “layer thickness vs. layer number” show cyclic trends in the variability of layer thickness across the sampling area (Figs. 12D–12E). Similar cyclic trends in anisotropic structures are also observed on Mars and Earth (Smolyar, Bromage & Wikelski, 2016).

Figure 12 Layer thickness variability across fragments of Martian surface with isotropic structure of layers.

(A) Layered structure of Martian surface. (B) First (green) sampling area. (C) Second (orange) sampling area. (D) Cyclicity in variability of layer thicknesses across 2D green sampling area. (E) Cyclicity in variability of layer thicknesses across 2D orange sampling area. Image credit: NASA/JPL/University of Arizona. PSP_006609_1330.

Materials. Lamellar/rippled/layered patterns have been found in metals, alloys, insulators, semiconductors, and many others materials (Deville, 2018; Zuo et al., 2016; Moya, 1995). Lamellar thickness is a micromorphological characteristic that plays a central role in the relationship between a material’s microstructure and its macro properties because “the unique properties of natural layered materials and nanocomposites are achieved through a fine control of the layer thickness” (Deville, Tomsia & Saiz, 2007, p. 970). Figure 13A depicts an image of layered Al–Si composite with anisotropic structure. The chart of “layer thickness vs. layer number” shows cyclicity in the variability of layer thickness across the sampling area (Fig. 13B), which is divided into parts a, b, c, and d (Fig. 13C) according to the uniform distribution of layer thickness in each part (Fig. 13D). It follows that the chart of “layer thickness vs. layer number” provides a more detailed description of a layered pattern’s morphological characteristic than average layer thickness. The signal-to-noise ratio (Smolyar, Bromage & Wikelski, 2016) for chart (Fig. 13B) is equal 6.

Figure 13 Layer thickness variability across a 2D plane: material.

(A) Sampling area of Al-Si material. (B) Variability of layer thickness across 2D sampling area. (C) Four subareas, a, b, c, and d, exhibit uniform variability of layer thickness. (D) Average layer thickness for regions a, b, c, and d. Image credit: Sylvain Deville.

Medicine. Figure 14A shows a PC (lamellar), a sensory receptor in skin that is sensitive to contact and vibration. The anisotropic lamellar structure of PCs plays an essential role in the function of the PC system; lamellar thickness and number of lamellae are used to examine the link between the PC’s material and morphological characteristics and its response to vibration (Quindlen et al., 2017). We use 42 transects to plot the chart of “layer thickness vs. layer number” (Fig. 14B), which clearly demonstrates the cyclic nature of variability in lamellar thickness across the sampling area.

Figure 14 Layer thickness variability across a 2D plane: Pacinian (lamellar) corpuscle.

(A) Sampling area of Pacinian corpuscle. (B) Variability of lamellar thickness across 2D sampling area. Image credit: Ed Uthman. Wikipedia contributor. “Lamellar corpuscle,” https://en.wikipedia.org/wiki/Lamellar_corpuscle#/media/File:Pacinian_Corpuscle_(36298105211).jpg, licensed under CC BY 2.0.

Birds. The chart of “layer thickness vs. layer number” exhibits non-random trends in the variability of layer thickness across bird feathers (Figs. 15A–15C). The chart and DStr could potentially serve as morphological characteristics of birds with application to the study of their life cycles. Striped patterns are often used to distinguish bird species from one another. In particular, shrikes and their relatives are recognizable to birders by the peculiar differences in the thickness and layering of their striped patterns, many of which are simply black and white. Furthermore, there are often marked differences in the morphological features of feathers between males and females of the same species, a dimorphism that is recognized both by the animals themselves and by human observers (Gluckman, 2014). The signal-to-noise ratio (Smolyar, Bromage & Wikelski, 2016) for chart (Fig. 15C) is equal 6.

Figure 15 Layer thickness variability across a 2D plane: banded pitta and owl feathers.

(A) Layered plumage pattern of banded pitta. (B) Layered plumage pattern of owl. (C) Non-chaotic variability of layer thickness across plumage patterns of banded pitta and owl. Image credit: (A) Doug Janson. Wikipedia contributor. “Javan banded pitta,” https://en.wikipedia.org/wiki/Javan_banded_pitta#/media/File:Pitta_guajana-20030531.jpg, licensed under CC BY SA 3.0. (B) Phillip Anderson.

Spider web. The morphology of orb (circular) spider webs (Fig. 16A) is frequently studied not only because of their superior mechanical properties but also as a source of information about spiders’ construction behaviors (Eberhard, 2014; Soler & Zaera, 2016). The orb web represents a layered system with structural anisotropy: “One of the most relevant structural traits of orb webs is their mesh width” (Zschokke & Nakata, 2015, p. 661). Mesh width (i.e., layer thickness) is used to understand the construction features of web systems and relate them to spiders’ behavior. For instance, Zschokke & Nakata (2015, p. 661) point out that “a closer look at the orb webs reveals that mesh widths are not the same throughout the entire web.” Charts describing the variability of mesh width across the sampling area (Fig. 16B) confirm this statement and indicate cyclicity in the variability of mesh width across the sampling area. Thus, EM could be used to generate a new set of structural characteristics describing the anisotropy of an orb web and its segments.

Figure 16 Layer thickness variability across a 2D plane: spider web.

(A) The segment of the spider web used for analysis. (B) Variability of layer thickness across 2D segment. Image credit: Samuel Zschokke.

Sensitivity of DStr to minor structural changes

Next, we examine how minor changes in layer structure affect DStr, using fingerprint (Fig. 17A), fish scale (Fig. 17B), and an eye angiogram (Fig. 17C) as test objects. Let us denote characteristics of images before and after structural changes by DStr(before changes) and DStr(after changes). We describe the link between DStr(before changes) and DStr(after changes) and structural changes in images in quantitative terms using the following procedure: First, we describe the difference between DStr(before changes) and DStr(after changes) on a relative scale (%). All changes in layer structure are marked in red. We denote the difference as Parameter-1. Second, we describe (in %) the difference (in pixels) between the images before and after changes. To do so, we calculate the number of black pixels in an image before changes (total pixels before changes) and the total number of pixels that change color (white to black or vice versa) as a result of structural changes (total pixel change). The ratio (%) of “total pixel change/total pixels before changes” allows us to calculate the magnitude of structural changes in an image. This ratio is denoted Parameter-2. The relation between Parameter-1 and Parameter-2 allows us to estimate the sensitivity of DStr to structural changes in the image. Table 1 comprises results of calculating Parameter-1 and Parameter-2. The average ratio between Parameter-1 and Parameter-2 is 1.31:0.1, which implies that a 1% structural change in layers results in a 13.1% change in DStr. This result provides evidence that minor changes in layer structure are accompanied by substantially greater changes in DStr values. This feature of EM could be positive or negative, as required by the application.

Figure 17 Sensitivity of parameter DStr to changes in layer structure.

(A) Sensitivity of DStr to minor changes in a fingerprint pattern structure. (B) Sensitivity of DStr to minor changes in a fish scale pattern structure. (C) Sensitivity of DStr to minor changes in an eye angiogram pattern structure.

Table 1 Comprises results of calculating Parameter-1 and Parameter-2.

Image	Parameter-1 (%)	Parameter-2 (%)	
Fingerprint	0.42	0.072	
Fish scale	3.00	0.150	
Eye angiogram	0.52	0.077	
Average	1.31	0.1	

Sensitivity of DStr to binarization of layered patterns

Let us examine the sensitivity of DStr to binarization of initial grayscale patterns since in present work we use grayscale patterns of varying quality and origins. We have chosen the surface of Mars (Figs. 18 and 19) and the human aorta (Fig. 20) for experiment. We chose outline, that is, contour trace and central line trace modes (Bouton, 2017) since they represent two distinct approaches for image binarization (i.e., the central line trace mode substantially changes the shape of layers with respect to outline trace (Files S3 and S4 comprised binary images in two modes)). The experiment consists of several steps:

Figure 18 Area of study on the Martian surface.

Image credit: NASA/JPL/University of Arizona. ESP_021737_1710_RED.

Figure 19 Coordinate system of the area of study on the Martian surface.

Image credit: NASA/JPL/University of Arizona. ESP_021737_1710_RED.

Figure 20 Area of study of a human aorta.

Image credit: Hans Snyder/JMD Histology & Histologic Inc.

Step 1. Grayscale image of the Martian surface after embossing (Fig. 18) are divided into 192 squares (Fig. 19) and the human aorta are divided into 62 squares (Fig. 20). An ID is assigned to each square (Figs. 19 and 20).

Step 2. We use two modes (contour trace and central line trace) for image binarization. Notation for file name is the following: image A-04.bmp represents contour trace square for ID = A-04, and image A-04-1.bmp represents central line trace square (File S3 for Mars and File S4 for human aorta).

Step 3. The parameter DStr is calculated for each Mars and human aorta square with application contour trace and central line trace modes, resulting with DStr(contour) and DStr(line), respectively. The total number of images in the experiment is 384 (Mars) + 124 (human aorta) = 508 (binary layered images).

Step 4. Charts DStr(contour) vs. DStr(line) is plotted for Mars (Fig. 21A) and human aorta (Fig. 21B). Raw data available in File S5.

Step 5. Frequency diagrams for Mars and human aorta are plotted (Fig. 21C).

Figure 21 shows that the degree of similarity between DStr(contour) and DStr(line) is very high for Mars and human aorta patterns since all points on the charts situated along straight lines with R2 > 0.965 for Mars (Fig. 21A) and R2 > 0.875 for human aorta (Fig. 21B). This indicates that there is no single square on Mars’s area of study with a high value of DStr(contour) and low level of DStr(line) and vice versa. This is also true for the image of the human aorta. Only for a few images do the differences between DStrMars(contour) and DStrMars(line) or DStraorta(contour) and DStraorta(line) exceed 3%; for more than 70% of the images, the difference does not exceed 2% (Fig. 21C). Thus, outline trace and central line trace functions generated binary images with close values of DStr for chosen patterns of Mars (Fig. 18) and the human aorta (Fig. 20). To study a particular area of Mars (Fig. 18), contour trace and central line trace could be used for binarization if we assume that quality of initial satellite image of Mars surface allows us to ignore differences of less than 4% between DStrMars(contour) and DStrMars(line). In the case of the human aorta, differences of less than 3% between DStraorta(contour) and DStraorta(line) could be unacceptable from a medical point of view.

It is necessary to point out that this experiment is not sufficient to allow us extend these assumptions to patterns of other categories such as lamellar bones, alloys, or layered systems of animals and plants. Moreover, it is difficult to predict the relationship between DStr(contour) and DStr(line), even for other areas of Mars surface or human aortae. Thus, the first step in applying the proposed method to solving any particular problem has to be studying the link between DStr and pattern binarization. The present work shows the applicability of the proposed methods to layer patterns of different categories but does not create any insight into any particular layered system or solve any specific problem. Thus, we use one version of the binarization protocol (Smolyar, Bromage & Wikelski, 2016).

DStr as a tool for detecting structural anomalies in layered patterns

Empirical models are generally used to predict, measure, and characterize an object under study (Clarke & Primo, 2012). In this section, we illustrate the usefulness of DStr by characterizing the distribution of structural anomalies in an image of layered landform on Mars. We chose the Martian surface as object of study since Mars dunes and ripples are “one of the leading factors in landscape evolution” (Chojnacki et al., 2019, p. 427). The morphology of dunes and ripples and their dynamics are indicators of variability in the Martian climatic system (Read, Lewis & Mulholland, 2015).

From an image of a Mars layered landform (Fig. 18) that is approximately 50 km2 in size, we would like to determine those regions with the greatest differences in layer structure. To solve this problem, we take the following steps:Divide the area of study into 192 squares of equal size (Fig. 19).

Calculate DStr for each square by averaging DStr for contour trace and central line trace modes.

Use a color scale to visualize the distribution of DStr across the area of study (Fig. 22A). Square M-18 has minimal structure (DStr = 0.148); square D-08 has maximal structural disorder (DStr = 0.443). Since DStr = 0.5 is the maximal value for layered structure (section Layered vs. non-layered systems), the difference in relative scale (%) between DStr(M-18) and DStr(D-08) is equal to 59%:

Figure 21 Sensitivity of DStr to contour and line trace modes of layered patterns binarization.

(A) Mars. Correspondence between two modes of image binarization: DStr(contour trace) and DStr(line trace). (B) Aorta. Correspondence between two modes of image binarization: DStr(contour trace) and DStr(line trace). (C) Statistics of differences between DStr(contour trace) and DStr(line trace) for images of Mars and aorta.

Figure 22 Distribution of DStr across layered Martian surface.

(A) DStr of 0.5 × 0.5 km grid system of Martian surface. (B) The region that comprises nearby grid cells with contrasting values of DStr. Image credit: NASA/JPL/University of Arizona. ESP_021737_1710_RED.

DStr(M-18) = 29.6%, DStr(D-08) = 88.6%, DStr(D-08) − DStr(M-18) = 59%.

Figure 22B shows the example of the region with contrasting values of DStr.

In summary, Fig. 22 justifies the use of DStr as a local morphological characteristic of a sizable layered landform. DStr allows us to reveal subregions with anomalous low or high DStr values. Pinpointing structural anomalies on sizable layered landforms is important for understanding climatic system of Mars (Fenton & Hayward, 2010; Banks et al., 2018) and for exploring candidate landing sites for the Mars 2020 rover (Chojnacki, Banks & Urso, 2018).

Discussion

Method: pros and cons

The present work introduces the idea of structural disorder in layered systems. The difference between DStr and the geometric morphometric approach to quantifying structures (Adams, Rohlf & Slice, 2004) is that DStr measures the deviation of anisotropic layer structures from isotropy.

As a measure of image disorder, entropy is broadly used in various image processing applications (Tsai, Lee & Matsuyama, 2007). Image entropy is calculated as the distribution of pixel values across a 2D image. DStr is also a measure of layered image disorder that is fundamentally different from image entropy; DStr is the measure of layered patterns anisotropy i.e., deviation from isotropic analog. The basis for calculating DStr is N-partite graph G(N).

Also, it is usual practice to choose parameters for describing patterns based on the specific characteristics of an object of study. DStr and charts of “layer thickness vs. layer number” and “layer area vs. layer number” can be used globally as well as locally to describe the morphological characteristics of any anisotropic layered pattern. This property of DStr and the charts allows us to formulate new questions, suggest new testable hypotheses about pattern formation, and identify links between properties and structures of study objects, extending areas of applications for analyzing various anisotropic layered systems. Experiments with aortas (Fig. 8), human hair (Fig. 10), and fingerprints (Fig. 11) illustrate that the parameter DGrp(R1, RN) (in addition to sampling density) complement DStr in quantifying the structural characteristics of layered systems.

Experiments with fingerprints (Fig. 11) illustrate that the chart for “DStr vs. number of transects,” which is highly accurate (R2 > 0.995), could be interpolated to a power function, y = mx−k. Because DStr is sensitive to minor structural changes (Fig. 17A) and the four basic categories of fingerprints have substantially different m and k parameters, y = mx− k could serve as a unique fingerprint ID.

Complicated layered patterns in bird plumage (Fig. 9A–9E) are formed by multiple individual feathers that have, individually, relatively simple patterns. Thus, it is not quite clear whether layer thickness is distributed chaotically across the body or demonstrates trends similar to other layered systems. We calculate DStr (Figs. 9F–9G) and the chart for “layer thickness vs. layer number” (Fig. 15C) in order to analyze the morphology of these layers. Because DStr(Pitta, Owl) <0.5, we conclude that the feather patterns form layers in these species. Charts for “layer thickness vs. layer number” (Fig. 15C) exhibit trends in variability of layer thickness across pitta and owl bodies. These results justify the potential applicability of DStr and the chart for “layer thickness vs. layer number” for describing morphological characteristics of bird plumage. In this context, it should be noted that the distinction between layer thickness and layer number is also used by birds themselves to distinguish among different species, even in a generalized way. Sparrow hawks, which are among the most fearsome predators of small birds, particularly songbirds, have a particular pattern of feathers on the breast plumage. Sparrow hawks are mobbed by small birds all over the world, and the feather patterns alone suffice to entice small birds to engage in the mobbing behavior.

The structure and size of lamellar bone form a record of the state of internal and external factors responsible for lamellar formation over an organism’s life history. The cyclicity of lamellae thickness (Bromage et al., 2009) over the period of formation is a cumulative effect of many cyclic factors. Many hard tissues form incremental patterns at varying time scales. For instance, mammalian enamel and dentine develop according to a circadian rhythm, creating a pattern visible as daily microanatomical growth lines. These tissues, as well as those of bone, have also been observed in some mammals to contain longer-period developmental rhythms that scale with body mass (Bromage et al., 2012). These hard tissue rhythms are of substantial interest in mammal life history research, providing information about the duration and amplitude of periodic phenomena as well as about other natural history events occurring during bone and tooth formation, which for some species could not be obtained by other means. It is also significant that hard tissue rhythms are often preserved after an organism’s death, either as resilient hard tissue or as a fossil. Incremental patterns are a primary source of information about the duration and amplitude of periodic phenomena as well as about other natural history events occurring during formation: Information about cyclicity, interactions between environmental and/or physiological cycles, and perturbations to the responding system are all inherently contained in these incremental patterns (Bromage et al., 2011).

The variable cyclicity of layer thickness across the sampling area of an anisotropic layered system is to be expected because “the whole pattern (of nature) is of cycles within cycles within cycles” (Medawar & Medawar, 1983, p. 73). EM = {BF, G(N), TM,N} provides tools to reveal cyclicity hidden in layered anisotropic environments.

Let us consider some of the limitations of the proposed method. Many limitations are as yet unknown because the morphology of anisotropic layered patterns is a relatively new object of study. Thus, we list the most obvious limitations that follow from the image analyses presented in the Results section:Images of the Martian surface exhibit layered patterns as a result of processes occurring in different space–time domains. The proposed method does not provide tools to describe global structural parameters of this category of images.

Many layered patterns consist of lines with simple shapes, but the images of the human aorta (Fig. 8A) and PC (Fig. 14A) have more complicated configurations. The proposed method ignores the shape of layers.

It is necessary to quantify the spatial orientation of layers when developing new materials (Deville, 2018) and setting up correspondence between the morphology of layered systems and water temperature (Olson et al., 2012; Gilbert et al., 2017). The proposed method does not provide tools to quantify the preferential orientation of layers.

The EM = {BF, G(N), TM,N} does not account for the material properties of layers.

All of the transect versions used to calculate DStr are plotted in one direction, which is perpendicular (or quasi-perpendicular) to the layers.

The problems of layered pattern normalization and alignment are outside the scope of this work.

Possible experimental tests

Many factors that contribute to the formation of layered patterns in living systems have cyclic natures. For instance, layers in growth systems are formed in direct response to cyclic environmental factors such as temperature (Goodwin et al., 2001; Izzo & Zydlewski, 2017), tides (Poulain et al., 2011), and light–dark rhythms (Scrutton, 1978; Smith, 2006). Cyclic planetary dynamics can also affect the formation of growth increments (Clark, 1974; Pannella & MacClintock, 1968; Kahn & Pompea, 1978; Vanyo & Awramik, 1985). On the surfaces of Earth and Mars, winds are mainly responsible for the formation of dunes and ripples (Lapotre et al., 2016; Kok et al., 2012). It is reasonable to suggest that the cyclicity of layer thickness stems from the cyclicity of factors controlling layer formation, but this explanation is not always possible. Layered patterns are the cumulative result of many factors occurring in different space–time domains, not all of which are cyclic, and not all factors are known.

Notwithstanding the fact that each object of study has unique properties, the layers of various systems are all formed in the gravity fields of the massive rotating bodies of Earth, Mars, and other planets. Thus, it would be reasonable to explore the influence of zero-gravity (i.e., extreme external factors on layer formation). A promising approach would be to examine the influence of extreme external factors—such as zero-gravity, extreme temperature, radioactive contamination, low oxygen, and absence of light—on layer morphology. EM is a suitable tool for such experiments since it is sensitive to minor structural changes (Fig. 17) and allows us to detect anomalies in layered systems (Figs. 6A and 6B).

Areas of application

Macro-, micro-, and nanostructures play a vital role in understanding pattern formation and relationships between processes and structures (Aizenberg & Fratzl, 2009). Central problems in studying layered objects—particularly in medicine (Novotny et al., 2017), materials science (Deville, 2018), and biomimetic research (Meyers, Hodge & Roeder, 2008; Gilbert et al., 2017)—are quantitatively describing the relationship between structure and properties, tracking structural changes over a period of time, and revealing structural anomalies. Experiments with various categories of layered systems justify the possibility of using EM = {BF, G(N), TM,N} to help to solve these problems.

Many layered objects—such as corals, fish scales, and bivalve shells—are formed in the world ocean. Morphological characteristics of growth increments in these objects are a function of seawater parameters and changes in the space–time domain (Carroll et al., 2014). EM could be used to analyze growth increments of shells in a seawater environment. Our interest in the link between growth increments and the marine environment is based on available marine data products, new instrumental technology measuring the chemical composition of seawater, and recently published discoveries of relationships between the morphology of shell growth increments and seawater temperature (Gilbert et al., 2017).

Marine data products. The World Ocean Database (WOD) and International Comprehensive Ocean-Atmosphere Data Sets (ICOADS) are the world’s largest freely available marine databases. WOD comprise 16+ million globally distributed profiles, beginning with instrumental observations in 1772 through the present (Levitus, 2012; Boyer et al., 2014). A profile is the set of measurements of physical, hydrochemical, and plankton characteristics of seawaters on the surface and at various depths.

International Comprehensive Ocean-Atmosphere Data Sets is an archive of global near-surface marine data, with over 456 million individual records since 1662 (Woodruff et al., 2011; Freeman et al., 2017). Each record is a set of sea-surface temperature and marine meteorological parameters such as wind speed and direction, humidity, sea-level pressure, cloud cover, sea state, sea ice, and descriptive information such as type and amount of cloud cover at different levels in the atmosphere. ICOADS and WOD are used to study local (Reagan, Seidov & Boyer, 2018; Seidov et al., 2017; Matishov et al., 2014; Ansell et al., 2006; Marullo, Artale & Santoleri, 2011) and global (Rayner et al., 2003; Garcia et al., 2005; Ishii et al., 2005) climatic characteristics of the world ocean and its dynamics.

Seawater temperature vs. growth increment morphology. Gilbert et al. (2017) developed a novel method that allows us to reconstruct present and past seawater temperature by analyzing the morphology of modern and fossil shells, which “complements the strength and compensates for the weaknesses of existing geochemical method” (p. 291). EM could be used to formalize some stages of layered image processing and account for the structural anisotropy of shells’ growth increments. WOD could be used to define areas of the world ocean suitable to examine the influence of seawater parameters of different water masses on the development of shells’ growth increments. Gilbert et al. (2017) hypothesized that factors such as salinity, pH, or nutrients can affect the morphology of shells’ growth increments in addition to water temperature. Within the frame of this hypothesis it would be reasonable to examine the influence of seawater chemical composition on the development of shells’ growth lines. The new method for measuring the chemical composition of fresh and saltwater (Bäuchle et al., 2018) could be used for this purpose.

Measuring periodic table in fresh and salt waters (Bäuchle et al., 2018). Water is an accumulation of dissolved elements in the form of organic (typically carbon-hydrogen-based) and inorganic (non-organic) molecules. Given the importance of water to all life, it is astonishing that not a single aqueous sample has ever been measured for element concentrations across the breadth of the chemical periodic table. This dearth of research is not for the lack of want for knowledge but because technologies for detecting all elements in a water sample have been unwieldy and expensive to operate. A recently developed “simultaneous Mass Spectrometer” ICP-MS (si-ICP-MS) permits 71 inorganic elements to be detected in one evaluation from small sample volumes in seconds and at relatively low consumable and personnel costs.

To examine the potential of si-ICP-MS for evaluating environmental water, and for assessing its usefulness in studies of incremental structures, we first measured tap, well, rain, freshwater lake, river, seawater, and snow. Figure 23 depicts the distribution of fresh and saltwater samples. Most of the periodic table is indeed represented in environmental water, which includes municipally treated tap water (Smolyar, Bromage & Wikelski, 2019: Fig. 33). This is fascinating because snow is essentially the same as all other fresh waters, which indicates that the atmosphere—after being scrubbed by snowflakes—is fundamental to the movement of elements at high latitudes and altitudes around the world. Seawater stands out as having higher abundances of elements overall.

Figure 23 Distribution of water samples.

The WOD, integrated with chemical composition of seawater, will thus allow us to examine the influence of a broad spectrum of seawater characteristics on the development of growth increments in marine life such as coral, fish scales, and shells. Additionally, the chemical composition of soil and air allows us to use EM to quantify the correspondence between environment and growth patterns of terrestrial plants and animals.

To appreciate the relevance of such data to the study of incremental structures, we examined the lamellar bone of a subsistence fisherman who lived around a freshwater lake. We used a laser ablation system attached to the si-ICP-MS to measure the same elements measured from the lake water on which he made his living. We have made two interesting observations from this research: First, the inorganic spectrum of elements in the local water and in a bone from the fisherman were quite similar.

Second, we discovered that the lamellar increments of bone are formed on the same interval at which the growth increments in enamel, the striae of Retzius, are formed (Bromage et al., 2009). Striae of Retzius may be calibrated in absolute time, and in this fisherman that period was 8 days. Roughly 15 years of continuously formed lamellar bone were available from years for which we have meteorological data. In the example shown in Fig. 24A, we demonstrate, for instance, that from 1981 to 1995, the concentration of Strontium (Sr) varies cyclically (Fig. 24B) in its ratio with Zinc (Zn).

Figure 24 Zn/Sr ratio over 15 years of lamella bone.

(A) Area of study (blue rectangle). (B) Zn/Sr ratio over 15 years of lamella bone.

Bird migration. Other examples in which marine data could relate to the structure of living organisms are birds’ morphology and their migration patterns (Shaffer et al., 2006) across the world ocean (Fig. 25). ICARUS, short for “International Cooperation for Animal Research Using Space,” is a global collaboration of animal scientists to establish a novel satellite-based infrastructure (Cooke et al., 2004; Wikelski et al., 2007) for Earth observation of small objects such as migratory birds, bats, or sea turtles (Pennisi, 2011). These findings will aid behavioral research, species protection, and research into the paths taken in the spread of infectious diseases. The information could even help predict ecological changes and natural disasters. In the process, ICARUS researchers will attach miniaturized transmitters to hundreds or thousands of animal species. These transmitters send measurement data via a code-division multiple access-encoded signal to a receiver station in space that transmits data to a ground station. The results will be published in a database that will be accessible to everyone at www.movebank.org. The ICARUS project provides us with tools to describe collective birds motion across the world ocean.

Figure 25 Sooty shearwaters migration routes across Pacific Ocean.

Image credit: Shaffer, et al. PNAS ©, August 22, 2006, 103.

A miniaturized, solar-powered animal tag can communicate with the ICARUS equipment at the International Space Station from a distance of up to 800 km, allowing it to record its absolute position at regular intervals using GPS and to acquire local temperatures, 3D acceleration, and 3D magnetometer values as well as pressure, altitude, and humidity, which give indications of the animal’s behavior, internal and external state, and environmental conditions—all using a tag with a mass less than five g and a volume of approximately two cm3.

Integrated morphological characteristics of individual birds, their migration routes, WOD and ICOADS, together with EM create a basis to formulate testable hypotheses of scientific and commercial value.

Conclusion

The key element of the present work is the notion of structural disorder in 2D layered systems (DStr), which is applicable to any layered object, irrespective of size or nature. Equation (3), which shows that layered patterns comprise anisotropic and IC components, provides a foundation for formalizing DStr. We intend in the future to investigate the applicability of Eq. (3) for processing categories of patterns beyond the layered patterns discussed here.

Various layered systems presented in this paper exhibit surprising levels of structural similarity, what Ball (2009, p. 177) called nature’s use of “not the Law of Pattern but a palette of principles.”

Supplemental Information

Supplemental Information 1 Excel file of raw data for calculating DStr = f(transect number) and DStr.

Click here for additional data file.

Supplemental Information 2 Binary patterns in the bmp format used in experiments (section Results).

Click here for additional data file.

Supplemental Information 3 Binary patterns in bmp format of the Martian surface in contour trace and central line trace modes.

Click here for additional data file.

Supplemental Information 4 Binary patterns in bmp format of the human aorta in contour trace and central line trace modes.

Click here for additional data file.

Supplemental Information 5 Raw data in the Excel format for calculation DStr for Martian surface and human aorta.

Click here for additional data file.

The authors thank Amy Bekkerman of Precision Edits for help in preparing the manuscript. We acknowledge G. Vovna and B. Ravvin for their discussions. The scientific results and conclusions, as well as any views or opinions expressed herein, are those of the author I. Smolyar and do not necessarily reflect the views of NOAA or the Department of Commerce.

Additional Information and Declarations

Competing Interests

Author Contributions

Patent Disclosures

Data Availability

The authors declare that they have no competing interests.

Igor Smolyar conceived and designed the experiments, performed the experiments, analyzed the data, contributed reagents/materials/analysis tools, prepared figures and/or tables, authored or reviewed drafts of the paper, approved the final draft.

Tim Bromage conceived and designed the experiments, performed the experiments, analyzed the data, contributed reagents/materials/analysis tools, prepared figures and/or tables, authored or reviewed drafts of the paper, approved the final draft.

Martin Wikelski conceived and designed the experiments, analyzed the data, contributed reagents/materials/analysis tools, prepared figures and/or tables, authored or reviewed drafts of the paper, approved the final draft.

The following patent dependencies were disclosed by the authors:

Smolyar, I.V. 2014. System and Method for Quantification of Size and Anisotropic Structure of Layered Patterns. U.S. Patent 8,755,578, issued June 17, 2014.

The following information was supplied regarding data availability:

Raw data is available in the Supplemental Files.

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
