# Peer review of "Layered patterns in nature, medicine, and materials: quantifying anisotropic structures and cyclicity"

_PeerJ, doi:10.7717/peerj.7813_

## Round 0.1 · original submission · Major Revisions

Your manuscript has now been seen by 2 reviewers. You will see from their comments below that while they find your work of interest, some major points are raised. We are interested in the possibility of publishing your study, but would like to consider your response to these concerns in the form of a revised manuscript before we make a final decision on publication. We therefore invite you to revise and resubmit your manuscript, taking into account the points raised. Please highlight all changes in the manuscript text file.

Reviewer 1 ·

Basic reporting

The article proposes a method to quantify the structure of patterns found in several natural images. The authors suggest an approach based on the binarization of 2D images. In the definition of the empirical model, the variables are quite loose; they all depends a lot on the binarization of the grayscale image. How Tm,n is measured? This issue is raised in the sensitivity of DStr section. However, the authors, at the end of the section, leave to the application to count it as positive or not. This problem of image binarization is very well treated by the image processing community and there are issues that do not depend on the application for example if the image contains noise or blur. This has to be dealt with independently of the application. The other variable in the equation also depends on that which make the DStr very sensitive. The image processing part in the Results section does not explain how this procedure is done, it refers to another article from the authors, here at least some explanation is necessary.


In line 159, reference should be presented when the authors describe different publications.

In line 239, how in Figure 4B has a maximum of 103 transects?? how this number was obtained?? By the image of the figure does not appear to have 103 transects. Also, there is an inconsistency in this figure the end lines do not meet the dot in the left figure.

In Figure 3 C the reference to the layers does not correlate with the figure A., For example, A1,1, A1,2, and A2,1 in C give in misleading idea from figure A. The caption also should be complete. This figure is also exactly the same as from another article from the same authors. I suggest changing to give another example from a different configuration.

Experimental design

The experiments were done is very few loose images. To validate the experiments, a lot more images should be used, I suggest a complete dataset and also a way to metrify the usefulness of the proposed metric. There is no comparison or clear applicability of the metric this should be more explored to argue proposal. For example, what would be the results compared to an entropic measure applied in the image?

Validity of the findings

Since the method depends a lot on the binarization of the image is very difficult to access the validity. Also, there was no comparison with other methods that measure disorder such as entropy.

Additional comments

Even though the authors describe possible uses in several different areas there was no definite example of clear use of the proposed disorder metric to gain insight in a single problem. The authors should investigate better this issue why is this metric necessary??

Reviewer 2 ·

Basic reporting

This work by Igor Smolyar et al, presents a method to quantify the morphological characteristics of layered patterns and apply it to several images. It proposes three approaches: 1) Calculation of the disorder index DStr, 2) evaluation of the “layer thickness vs. layer number " graph and 3) evaluation of the" layer area vs. layer number" graph.
Although the three methods are described in the abstract, the authors do not present results for the 3rd.
It is definitely a judicious and interesting proposal for systems analysis made up of layers. The article is structured properly and the tables and figures are presented satisfactorily with only minor suggestions for changes.
It would be interesting to include the scale of the images used.
Equations can be best presented using equation editors.
The Sensitivity of DStr results, can be shown in a table. It is easier to refer.
I think it is interesting to present, for all applications, the Black & White image, used for layer evaluation.

Experimental design

no comment

Validity of the findings

The DStr sensitivity analysis results presented in the paper, shows that the method can present a high sensitivity to noise, reaching the ratio 30:1.5 (1.5% of change in the image pixels results in 30% of change in the index). This sensitivity is also highly dependent on layer structure, as can be seen when the Fish Scale and Fingerprint results are compered. It would be interesting to evaluate whether the sensitivity depends on the absolute value of the DStr. This can be easily done with a graph showing the DStr values in X and the Sensitivity in Y for multiple images examples.
This sensitivity may also imply spurious results depending on the image conversion parameters for B & W. It would be interesting to present how the image conversion parameters can influence the DStr estimation.
The authors present the fluctuation patterns in the thicknesses of the layers, however no evaluation is made on the characterization of these fluctuations. A simple evaluation of the autocorrelation or persistence (Hurst exponent) of the series could reveal interesting characteristics of the formation mechanisms of the layers.

---

## Round 0.2 · Minor Revisions

Your manuscript has now been re-reviewed. Although we are interested in the possibility of publishing your study, some concerns still need to be addressed before we make a final decision on publication. We therefore invite you to revise and resubmit your manuscript, taking into account the points raised. Please highlight all changes in the manuscript text file.

Reviewer 1 ·

Basic reporting

The authors present a revised version of the article following the reviews comments. Even though the authors improved the description of the method, the binarization process in which the grayscale image is transformed into a binary image is not explained the authors used the reference Smolyar (2016), which it seems that uses a simple threshold method. This is of importance since the result of the threshold defines the binary image and consequently the EM and DStr.

Experimental design

no comment

Validity of the findings

I would agree with the review D that it is necessary to show clear use of the proposed metric. Stating that this is out of the scope of the article is not a good argument for that since then why is this metric important or necessary? The authors should present a more convincing argument for the necessity of the proposed metric as in the conclusion it is stated that it could be potentially used to quantify the morphological characteristics of arbitrary 2-D binary patterns. If so how?

---

## Round 0.3 · accepted · Accept

We are delighted to accept your manuscript for publication.